# Senescent Secretome of Blind Mole Rat *Spalax* Inhibits Malignant Behavior of Human Breast Cancer Cells Triggering Bystander Senescence and Targeting Inflammatory Response

**DOI:** 10.3390/ijms24065132

**Published:** 2023-03-07

**Authors:** Amani Odeh, Hossam Eddini, Lujain Shawasha, Anastasia Chaban, Aaron Avivi, Imad Shams, Irena Manov

**Affiliations:** 1Department of Evolutionary and Environmental Biology, Faculty of Natural Sciences, University of Haifa, 199 Aba Khoushy Avenue, Mount Carmel, Haifa 3498838, Israel; 2Institute of Evolution, University of Haifa, 199 Aba Khoushy Avenue, Haifa 3498838, Israel

**Keywords:** *Spalax* fibroblasts, breast cancer cells, paracrine senescence, senescence-associated secretory phenotype (SASP), nuclear factor κB (NF-κB), interleukin 1 alpha (IL1α)

## Abstract

Subterranean blind mole rat, *Spalax*, has developed strategies to withstand cancer by maintaining genome stability and suppressing the inflammatory response. *Spalax* cells undergo senescence without the acquisition of senescence-associated secretory phenotype (SASP) in its canonical form, namely, it lacks the main inflammatory mediators. Since senescence can propagate through paracrine factors, we hypothesize that conditioned medium (CM) from senescent *Spalax* fibroblasts can transmit the senescent phenotype to cancer cells without inducing an inflammatory response, thereby suppressing malignant behavior. To address this issue, we investigated the effect of CMs of *Spalax* senescent fibroblasts on the proliferation, migration, and secretory profile in MDA-MB-231 and MCF-7 human breast cancer cells. The results suggest that *Spalax* CM induced senescence in cancer cells, as evidenced by increased senescence-associated beta-galactosidase (SA-β-Gal) activity, growth suppression and overexpression of senescence-related *p53*/*p21* genes. Contemporaneously, *Spalax* CM suppressed the secretion of the main inflammatory factors in cancer cells and decreased their migration. In contrast, human CM, while causing a slight increase in SA-β-Gal activity in MDA-MB-231 cells, did not decrease proliferation, inflammatory response, and cancer cell migration. Dysregulation of IL-1α under the influence of *Spalax* CM, especially the decrease in the level of membrane-bound IL1-α, plays an important role in suppressing inflammatory secretion in cancer cells, which in turn leads to inhibition of cancer cell migration. Overcoming of SASP in tumor cells in response to paracrine factors of senescent microenvironment or anti-cancer drugs represents a promising senotherapeutic strategy in cancer treatment.

## 1. Introduction

Cellular senescence is a stress response program triggered by a variety of cellular stressors, including telomere shortening, DNA damaging agents, oxidative stress, and the activation of oncogenes. Such cells halt division, which blocks the transmission of mutations to the next generation of cells limiting the propagation of potential oncogenic cells [1,2]. Cellular senescence is accompanied by the secretion of various molecules such as cytokines, chemokines, growth factors, matrix metalloproteinases (MMPs), collectively referred to as the senescence-associated secretory phenotype (SASP) [3,4]. The SASP factors can affect the surrounding tissues, and this effect can either promote or suppress tumor development. The modulating effect of SASP on cancer progression depends on the composition of the aging secretome (context dependence), the origin of the tumor, the state of the surrounding tissue, and host factors (such as biological age, hereditary predisposition, metabolism, and immunity) [5,6,7]. Both cancer and senescent cells share many common features such as genomic instability, epigenetic changes, telomere attrition (but unlike senescent cells, cancer cells overcome cell cycle arrest by activating telomerase) [8]. The secretion of cancer cells, depending on the type of tumor, contains a set of factors that are similar to the SASP of senescent cells (*IL6*, *IL1-ß*, *TGF-β*, *VEGF*, *MMPs* and others), which, in turn, activate normal stromal cells (fibroblasts, macrophages, etc.), thereby enhancing positive feedback loop and favors malignancy. In addition, oxidative stress and hypoxia in the niche of cancer progression can cause senescence and SASP in noncancerous cells. Thus, the outcome of a tumor is determined by the interaction of cancer cells and stromal cells of the cancer microenvironment, as well as the mutual influence of their paracrine secretomes. Normal fibroblasts are the main stromal compartment in the niche of cancer, and therefore their physiological state determines to a large extent whether or not the host microenvironment supports the cancer progression. Our previous studies have demonstrated the unique ability of the stromal fibroblasts of long-lived, cancer-resistant underground rodents, *Spalax*, to inhibit growth of various cancer cells either via direct cell-to-cell contact or paracrine secretion [9]. Different molecular design strategies that evolved in *Spalax* to cope with its extreme habitat (hypoxia and hypercapnia) were previously investigated in our laboratory [10,11]. Along with adaptation to environmental stress, *Spalax* has also acquired the ability to resist cancer. *Spalax’s* efficient DNA repair mechanisms maintain genome stability, which allows rapid repair of DNA damage [12]. In addition, among molecular strategies against cancer we found reduced capacity of adipose-derived stem cells to penetrate tumor, which leads to suppression of intratumoral abnormal angiogenesis [13]. The latest report from our laboratory showed that the secretory phenotypes of senescent *Spalax* fibroblasts and senescent tissues are devoid of major inflammatory factors. *Spalax* fibroblasts undergo replicative and etoposide-induced senescence, showing arrest of proliferation, SA-β-Gal-positive staining, and increased expression of *p21* and *p53*; however, the secretion of these cells lacked the expression of canonical inflammatory factors such as interleukins *IL6*, *IL8*, *IL1α*, *SerpinB2*, *GROα*, *ICAM-1* [14].

Senescent cells communicate with their environment via transmitting senescence phenotype to neighboring cells either normal or malignant through paracrine signaling of SASP factors [15,16,17]. Therefore, SASP determines how the senescent cells influence the tissue microenvironment: (1) SASP can induce pro-inflammatory microenvironment that attracts immune cells which are responsible for the clearance of damaged and senescent cells (physiological response that occurs, for example, during wound healing); (2) In aging tissues, immune processes are weakened, therefore, senescent cells are accumulated in tissues and able to transmit senescence to adjacent cells, giving rise to the release of inflammatory mediators. Persistent chronic inflammation (“inflammaging”) predisposes the development of destructive diseases associated with aging and supports the proliferation and invasion of precancerous and malignant lesions. Since *Spalax* resists oncogenic stimuli and maintains a non-permissive tumor environment, while *Spalax*’s cellular senescence is not accompanied by the secretion of major inflammatory mediators, we hypothesized that such a non-canonical secretome could sensitize cancer cells to senesce without triggering inflammatory SASP. Thus, the paracrine network that is activated in recipient cancer cells (bystander senescent cells) can be equivalent to that in the donor cells of senescent fibroblasts of *Spalax*. The present study aims to answer the following questions: (i) whether *Spalax* senescent secretome can induce senescence in cancer cells, lacking the acquisition of pronounced inflammatory SASP? (ii) whether non-canonical SASP of *Spalax* senescent fibroblasts can influence cancer cells throughout the reduction of their invasive and migratory behavior? (iii) to explore the role of IL-1α/NF-κB pathway in the regulation of inflammatory secretion and malignant behavior in cancer cells exposed to senescent *Spalax* secretome. To address these issues, we investigated the effects of conditioned medium (CM) of *Spalax* and human senescent fibroblasts on the proliferation, migration, and inflammatory response of MDA-MB-231 and MCF-7 human breast cancer cells. MDA-MB-231 is an extremely aggressive cell line producing a large number of inflammatory mediators that support high growth rates and metastatic capacity [18]. Therefore, the search for mechanisms that ensure the suppression of the inflammatory response during therapy-induced senescence may be a promising strategy for the treatment of such aggressive tumors.

## 2. Results

### 2.1. Spalax Senescent Secretome Decreases Proliferation and Induces Senescence-Associated Beta-Galactosidase (SA-β-Gal) Activity in MDA-MB-231 Cells

To investigate whether CM collected from *Spalax*, and human senescent fibroblasts could affect growth and induce senescence in MDA-MB-231 cells, the population doubling rate, cell cycle analysis and SA-β-Gal staining were performed (Figure 1). As demonstrated, a marked reduction in the proliferation of cancer cells treated with *Spalax* CM was observed after 48 h. Following 96 h of exposure, the cancer cell growth decreased 1.82 times compared to untreated cells (*p* ≤ 0.05). *Spalax* CM-treated MDA-MB-231 cells multiplied 16-fold within 96 h, while untreated and human CM-treated cells multiplied 29 and 28- fold, respectively (Figure 1A). Cell cycle analysis demonstrates induction of S and G2/M phase arrest in MDA-MB-231cells when treated with *Spalax* CM. No changes vs. control were found in MDA-MB-231 cell cycle distribution upon treatment of human CM (Figure 1B). Next, we evaluated whether MDA-MB-231 exposed to the senescent *Spalax* secretome undergo senescence. The level of SA-β-Gal activity, a well-known senescence marker, was significantly increased in cancer cells treated with *Spalax* CM when compared to control and human CM (*p* ≤ 0.001; Figure 1C,D). Subsequently, we questioned whether the senescence phenotype preserved in MDA-MB-231 cells during long-term cultivation under the influence of *Spalax* senescent secretome. As shown in Appendix A, *Spalax* CM significantly inhibited cancer cell proliferation (Appendix A), while the vast majority of cells retain the senescent phenotype after 10-day exposure (Appendix A)

### 2.2. Spalax CM Increases the Expression of Senescence Related p21 and p53 Genes, Enhances Nuclear Accumulation of p53 and Reduces the Levels of Spontaneous Double Strand Breaks in MDA-MB-231 Cells

The results presented in Figure 2A,B demonstrate that *Spalax* CM upregulates the expression of *p21* and *p53* mRNA levels in MDA-MB-231 cells. Human CM induces a moderate increase in the *p53* mRNA level in MDA-MB-231 cells. Appendix A shows a clear accumulation of *p53* in the nucleus of cancer cells treated with *Spalax* senescent secretome. MDA-MB-231 develops spontaneous DNA double strand breaks (DSB) as evidenced by the phosphorylation of γH2AX, a well-known marker of DSBs. Since *Spalax* fibroblasts developed a high DNA repair capacity [12], it was important to check whether CM collected from *Spalax* senescent fibroblasts could influence DSBs levels in MDA-MB-231 cancer cells. As shown in Figure 2C,D, CM harvested from human senescent fibroblasts increased the level of γH2AX foci in MDA-MB-231 cells, the effect being manifested already after 24 h of treatment and persisting after 96 h. Treatment with *Spalax* CM for 24 h did not alter the DSB level in MDA-MB-231 cells, whereas it decreased significantly after 96 h exposure (*p* ≤ 0.01) compared to control. Conversely, treatment of MDA-MB-231 cells with CM of senescent human fibroblasts resulted in an increase in DSB levels in cancer cells, presumably as a consequence of the large amount of SASP factors that are normally present in the CM of senescent fibroblasts in most mammals, excluding of the previously described phenomenon of “non-canonical SASP” in *Spalax*, whose cellular senescence and body aging is not accompanied by the secretion of major inflammatory factors [14].

**Figure 1 ijms-24-05132-f001:**
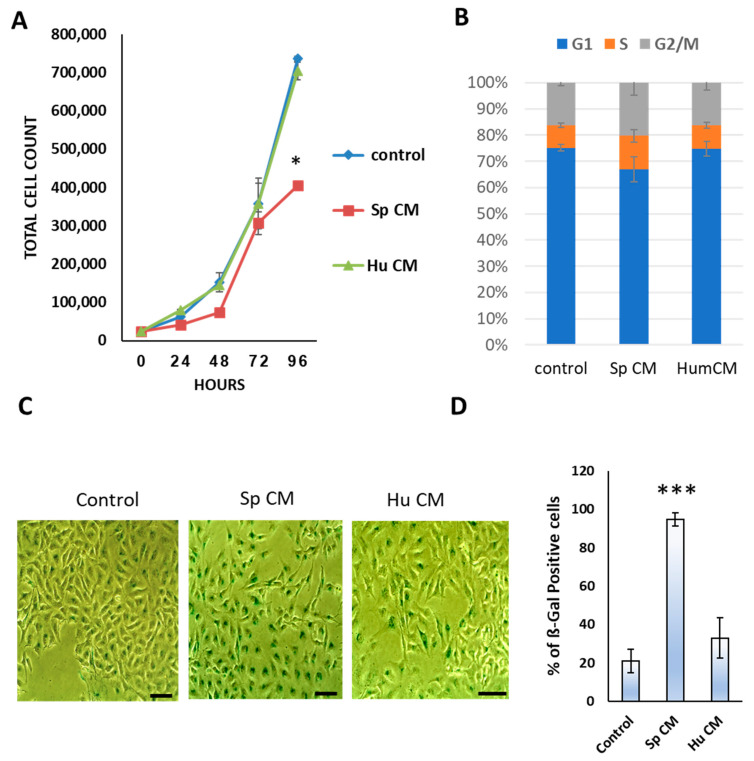
Influence of *Spalax* and human CM on proliferation rate and on the level of SA-β-Gal activity in MDA-MB-231 cells. (**A**) 25 × 10^3^ cells were seeded in 12-well plates under either *Spalax* (Sp) or human (Hu) CM, and counted after 24, 48, 72- and 96- hours. Values represent the averages of three independent experiments (n = 3) in triplicates. * *p* ≤ 0.05 differences between treated MDA-MB-231 with *Spalax* CM and control at 96 h (**B**) Flow cytometry analysis showing the percentage of cells in cell cycle stages. Values represent the average of two independent experiments in triplicate. (**C**) SA-β-Gal staining representative images: MDA-MB-231 cells were treated with *Spalax* and human CMs for 96 h. Bars, 100 μm (**D**) Percentage of SA-β-Gal-positive cells calculated from at least 300 cells in four independent fields for each biological repeat (n = 3) in triplicates (*Spalax* CMs were collected from senescent cells of three independent individuals). Human CMs were obtained from the same cells thawed at different times and at different passages (# 45–52; passages when human fibroblasts became senescent); *** *p* < 0.001 differences between control (untreated) and treated with *Spalax* CM.

**Figure 2 ijms-24-05132-f002:**
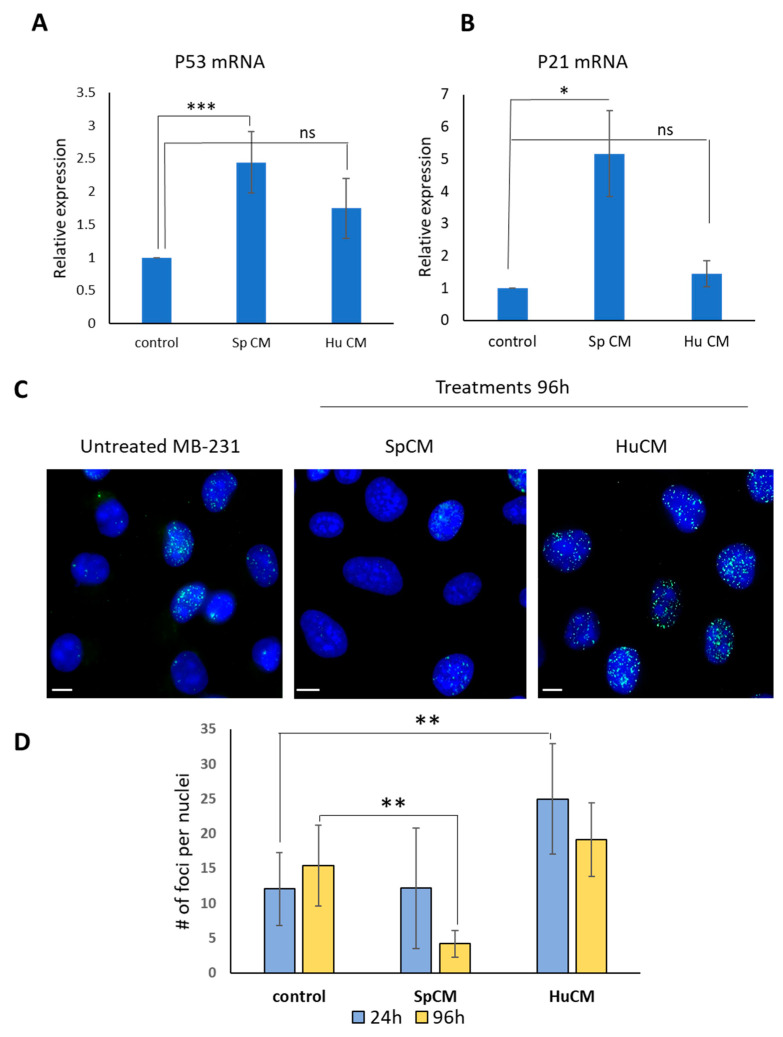
Effects of *Spalax* and human CMs on *p53/p21* senescence markers and on the spontaneous DSBs in MDA-MB231 cells. (**A**,**B**) *p21* and *p53* mRNA levels were quantified in MDA-MB231 under exposure to *Spalax* (Sp)/human (Hu) CMs by using relative qRT-PCR;. Experiments were performed in triplicates and repeated 3 times. Data are presented as mean ± SE * *p* ≤ 0.05; *** *p* ≤ 0.001, differences between treated MDA-MB-231 cells vs. untreated; “ns”, not significant *p* value. (**C**) MDA-MB231cells were treated with *Spalax* and human CM for 24- or 96 h; thereafter, cells were fixed and stained with anti- γ-H2AX antibody and counterstained with DAPI. Representative images of γ-H2AX (pSer13) foci in the nuclei of MDA-MB-231 untreated and treated for 96 h are shown. Bars, 10 μm. (**D**) The images were used for quantification using FociCounter software (minimum 250 nuclei per sample were analyzed). Data is presented as mean ± SD of three independent experiments (n = 3), ** *p* ≤ 0.01 differences between treated MDA-MB-231 with *Spalax* CM and control at 96 h. ** *p* ≤ 0.01 differences between treated MDA-MB-231 with human CM and control at 24 h. # means the number.

### 2.3. Spalax Senescent Secretome Reduces MDA-MB-231 Cancer Cells Migration

We were interested to test whether the senescent secretome of *Spalax* affects cancer cells migration. For these purposes, ‘scratch’ and Transwell^®^ migration assays were performed (Figure 3). *Spalax* CM diminished the scratch wound closure of cancer cells in a time dependent manner compared with control (untreated cells), whereas human CM moderately increased wound closure at 12 h and resulted in nearly complete closure at 24 h thereby enhancing cancer cell migration, once more at the expense of inflammatory factors that are present in the human CM (Figure 3A,B). Next, the effects of *Spalax*/human CMs on cancer cell migration were examined using Transwell^®^ migration chamber assay (Figure 3C,D). The results showed a significant decrease in the number of migrated cancer cells in the presence of *Spalax* CM compared to control cells, while human CM showed no effect on the ability of MDA-MB-231 cells to migrate.

### 2.4. CM of Spalax Senescent Fibroblasts Reduces the Levels of Inflammatory Factors in Secretome of MDA-MB-231 Cells

Based on the data presented above, the senescent secretome of *Spalax* fibroblasts causes senescence in MDA-MB-231 cells and decreases migration of cancer cells. Then we thought whether the secretory phenotype of cancer cells, which provides a high growth rate and metastatic capacity in an autocrine manner, changes under exposure to *Spalax* senescent secretome. In theory, such cells, while they senesce, should produce SASP factors even more intensively, and this is the main obstacle to therapy-induced senescence. However, as demonstrated above, the paracrine secretion of senescent *Spalax* cells, which triggers the senescence program in cancer cells, simultaneously reduces the level of DNA damage that, in turn, could inhibit NF-κB activation and the inflammatory response in these cells.

Indeed, exposure of MDA-MB-231 cells to *Spalax* CM reduced the expression of mRNA level of *IL6*, *IL8*, *COX-2*, *ICAM-1* and *GRO-α*, while treatment with human secretome resulted in upregulation of mRNA levels of these inflammatory mediators (Figure 4A). Next, we investigated whether secretion of inflammatory proteins IL6 and IL8, which are highly expressed in MDA-MB-231 cells, may be affected by *Spalax* CM. As demonstrated (Figure 4B), *Spalax* senescent secretome reduced secretion of IL6 and IL8, while human CM increased or did not change secretion of IL6 and IL8, respectively, when compared with control. Next, we examined whether the inhibition of the inflammatory response in MDA-MB-231 cells persists after prolonged exposure to *Spalax* senescent secretome. Since cancer cells are known to be able to emerge from proliferation arrest [19], we first tested the expression levels of *p53* and *p21* in MDA-MB-231 cells after long-term exposure to *Spalax* CM. Appendix A shows increased expression of *p21/p53* mRNA in MDA-MB-231 cells treated with *Spalax* CM for 10 days. This finding is consistent with reduced cell proliferation and positive β-Gal staining shown in Appendix A and confirms that cells remain in a senescent state. In parallel, the expression of SASP factors remains suppressed in these cells (Appendix A).

### 2.5. Spalax CM Suppresses the Expression/Phosphorylation of a Large Number of Signaling Proteins Associated with Inflammation, Proliferation and Migration in MDA-MB-231 Cells

The transcription factor NF-κB regulates a wide range of processes associated with inflammatory responses and cell proliferation. To reveal the NF-κB protein expression/phosphorylation profiles and status of related signaling molecules in cells treated with *Spalax* CM, NF-κB network antibody array was applied (Figure 5A,B). Of the 215 proteins analyzed, 123 were significantly suppressed, 84 did not undergo significant changes, and only six were enhanced in the treated cells compared to control. Among the downregulated proteins, NF-κB (p65; p105/50), IkB α/β, MAPKp38, CBP/p300 are widely represented (which are associated with the regulation and the secretion of inflammatory mediators). The level of expression and phosphorylation of phospholipase C-gamma 1 (PLCG1), which is involved in cell migration and metastasis of cancer cells, significantly decreased. In addition, protein kinase B (PKB or AKT) and protein kinase C (PKCA and PKCB), which are responsible for cell proliferation, migration and apoptosis, were also downregulated. Volcano plot (Figure 5B) displays the statistical significance of the differences (relative to the magnitude of difference) for every individual protein in the groups of MDA-MB-231 cells treated with *Spalax* CM versus untreated. Most NF-κB related proteins are significantly downregulated in treated cells versus control. Data points closer to “0” representing proteins that have similar expression levels (empty points). Western blots data confirm the suppression of phosphorylation of two of the most important players in the inflammatory response, namely p65 and p38, in MDA-MB-231 cells exposed to *Spalax* CM (Figure 6C,D and Appendix A).

### 2.6. The Role of IL1α in Suppressing the Inflammatory Response and Inhibition of MDA-MB-231 Cell Migration under the Influence of the Senescent Spalax Secretome

Treatment with *Spalax* CM resulted in redistribution of IL1-α compared to control, where IL1-α was mainly concentrated in the cytoplasm of cells near the nuclei (Figure 6A). Treatment with human CM did not lead to visible changes in the distribution of IL1-α within cells (arrows indicate the presence of IL1-α on the membrane). Flow cytometry confirmed a decrease of surface membrane-bound IL1-α in MDA-MB-231 treated with *Spalax* CM compared with non-treated cells or treated with human CM (Figure 6B). It is noteworthy that the effect of *Spalax* CM was weakened by the simultaneous treatment with the recombinant human IL-1α protein (IL-1α agonist).

**Figure 5 ijms-24-05132-f005:**
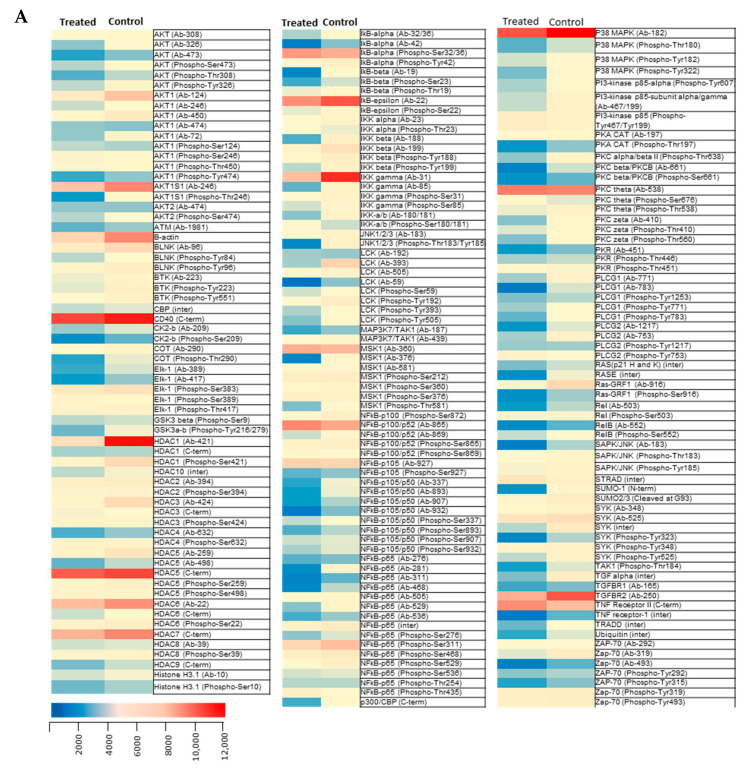
(**A**) Heatmap representing differences in the levels of expression/phosphorylation of NF-κB—related proteins in MDA-MB-231 cells exposed to *Spalax* CM (Treated) compared to control; (**B**) Volcano plot representing the log2 FC (treatment/control) versus −log10 *p* value (cut-off range—*p* value ≤ 0.05). Dots represent individual proteins.

We then examined whether a decrease in membrane-bound IL-1α affects NF-κB p65 and p38 phosphorylation in MDA-MB-231 cells exposed to *Spalax* CM. Western blot analysis shows that treatment by *Spalax* CM resulted in down-regulation of phosphorylation of both NF-κB-p65 and p38, but co-treatment with recombinant IL-1α agonist restores the level of NF-κB-p65 and p38 phosphorylation (Figure 6C,D and Appendix A). Treatment with human CM increased the level of phosphorylation of these proteins.

**Figure 6 ijms-24-05132-f006:**
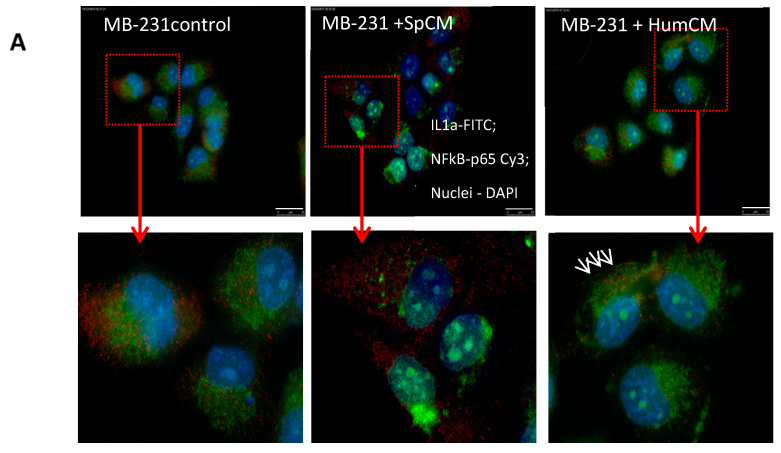
*Spalax* CM suppresses membrane-bound IL1-α and inhibits phosphorylation of NF-κB and p38 in MDA-MB 231 cells. (**A**) Representative images showing MDA-MB231 untreated and treated with *Spalax* (Sp) and human (Hum) CMs. A magnified view of the enclosed regions is shown on a low panel, highlighting the differences in IL1-α distribution within the cells between treated with human and *Spalax* CMs for 24 h and untreated. Treated/untreated cells were stained with IL1α–FITC and NF-κB-p65 antibodies; nuclei were counterstained with DAPI. Arrows indicate the presence of IL1-α on the membrane (**B**) Representative images of flow cytometry showing differences in IL1α surface membrane in MDA-MB-231 untreated, treated with *Spalax* CM, human CM, with or without recombinant human IL-1α protein. (**C**) Representative western blot analysis demonstrating the phosphorylation of p65 (Ser536), p38 in MDA-MB231 treated with recombinant IL-1α, *Spalax* CM (SpCM), *Spalax* CM+IL-1α (SpCM+IL-1α), and human CM (Hu CM). (**D**) Densitometry quantification of western blots. Western blots and densitometries of additional biological repeats (MDA-MB-231 treated with *Spalax* CMs generated from fibroblasts of different *Spalax* individuals) are presented in Appendix A.

### 2.7. The IL1α Agonist Promotes Cancer Cell Migration Weakened by Spalax CM through SASP Activation

We then analyzed how dysregulation of IL-1α induced by *Spalax* CM affects cancer cell migration. As demonstrated in Figure 3, *Spalax* CM suppressed wound healing and reduced the MDA-MB-231 migration. However, co-treatment of *Spalax* CM with recombinant IL-1α agonist diminished the effect of *Spalax* CM and restored MDA-MB-231 cancer cells migration (Figure 7A,B). Neither treatment by human CM nor human CM plus IL-1α agonist affected cancer cell migration. Since the IL-1α agonist activates NF-κB p65, a major regulator of the inflammatory response, we then examined the levels of SASP expression/secretion in MDA-MB-231 cells simultaneously exposed to *Spalax* CM and IL-1α. As shown (Figure 7C), when MDA-MB-231 cells exposed to *Spalax* CMs were simultaneously treated with IL1α agonist, the levels of *IL6* mRNA expression and IL6 protein secretion were increased.

### 2.8. Spalax Senescent Secretome Induces Senescence of MCF-7 Breast Cancer Cells, Inhibits the Secretion of Inflammatory Mediators and Hold Back the Migration of Cancer Cells

To prove that the observed anti-cancer effect is not limited to MDA-MB-231 cells, we also examined the acquisition of senescence in MCF-7 cells exposed to *Spalax* CM and wondered whether MCF-7 cell migration would be slowed down by the paracrine factors of *Spalax* senescent secretome. As demonstrated in Figure 8A, the level of SA-β-Gal activity was significantly increased in MCF-7 cancer cells treated with *Spalax* CM for 4 days when compared to control. In parallel, *Spalax* CM significantly reduced the migratory potential of MCF-7 cells, which resulted in slower wound closure by cancer cells exposed to *Spalax* CM compared to untreated cells (Figure 8B). As in the MDA-MB-231 study, *Spalax* CM reduced the mRNA expression level of a number of inflammatory mediators in MCF-7 cells (Figure 8C).

## 3. Discussion

We demonstrated here that the senescent secretome of *Spalax* fibroblasts, which we previously reported, lacking the major inflammatory SASP factors [14], is capable of transmitting senescence phenotype to cancer cells. Having become senescent, breast cancer cells MDA-MB-231 and MCF-7 replicate the SASP pattern of senescent *Spalax* cells, namely, they demonstrate a decrease in the secretion of inflammatory mediators, which, in turn, leads to the reduction of cancer cell migration.

Accumulating evidence suggests that the senescent phenotype can be transmitted from one cell to another via direct contact or via paracrine factors including extracellular vesicles [15,16,17,20]. Since cellular senescence is usually accompanied by an increased secretion of inflammatory factors, the recipient cell also becomes SASP positive. SASP, in its canonical pro-inflammatory and pro-growth design, is a barrier to therapy-induced senescence, which is a promising new cancer treatment strategy [21]: (i) cancer cells, in which senescence is induced, increase the production of their own inflammatory factors by triggering the SASP genes. Such cancer cells can transmit aging to neighboring cells, both neoplastic and non-malignant stromal cells; (ii) senescent stromal cells (the vast majority of which are fibroblasts, but also macrophages, immune cells, mesenchymal stem cells), in turn, release SASP factors. This reciprocal paracrine exchange gradually creates a tumor permissive microenvironment accelerating cancer cells proliferation, migration and stemness [22]. In this context, the tumor acquires resistance to anti-cancer drugs [23]. The deleterious effects of SASP can be abrogated by a senolytic strategy aimed at removing senescent cells. However, since senescent cells are involved in embryogenesis and wound healing, the widespread removal of senescent cells can disrupt these processes [21,24,25]. Unlike senolytics, the senomorphic strategy is designed to suppress SASP while maintaining cell cycle cessation in tumor cells [reviewed in [26]].

The composition of secreted molecules is extremely diverse and depends on the type of senescent cells, the type of stimulus that caused senescence, time passed since senescence initiation, the microenvironment, etc. [27,28]. However, a number of molecules are constant constituents among the various canonical SASPs, namely IL-1 α/β, IL-6, IL-8, GROα/β, MMP-1,3, MMP-10 ICAM-1, PAI-1 and IGFBPs [5]. Our recent research has shown that there are exceptions to this rule [14]. The secretory phenotype of replicative or etoposide-induced senescent fibroblasts of *Spalax* either did not contain many of the pro-inflammatory SASP factors (IL6, IL8, IL1α, GROα, SerpinB2, Cox2), or their expression reduced compared to young cells (ICAM-1). Decreased expression of pro-inflammatory SASP genes have also been found in senescent *Spalax* tissues. This SASP, devoid of major inflammatory factors, has been termed a “non-canonical SASP” [14]. 

In this report we demonstrated that Senescent secretome of *Spalax* induced senescence in MDA-MB-231 and MCF-7, as evidenced by increased SA- β-Gal activity, growth suppression and overexpression of senescence-related *p53/p21* genes. In parallel, we demonstrated accumulation of p53 in the nuclei of MDA-MB-231 cells exposed to *Spalax* CM, which probably reflects the release of p53 from targeted degradation and, as a result, slowing down/stopping cell division [29]. In addition, *Spalax* CM suppressed the level of spontaneous DSB in MDA-MB-231. Cellular senescence is typically driven by a persistent DNA damage response [30]. Double strand breaks in cancer cells occur spontaneously, a phenomenon termed ‘self-inflicted DNA DSBs’ which sustain tumorigenesis [31] and maintain the expression/secretion of main inflammatory factors (IL6, IL8, GROα, ICAM-1, Cox-2). Thus, cancer cells acquired a senescent phenotype along with the non-canonical SASP of senescent *Spalax* fibroblasts via paracrine factors. Thus, if in the previous report we demonstrated that the cessation of division of senescent *Spalax* cells was not accompanied by the secretion of the main SASP factors [14], here we received clear evidence that this type of senescence can be transmitted to the recipient cancer cells, which leads to a decrease in their aggressive behavior. Importantly, long-term exposure of MDA-MB-231 cells to *Spalax* senescent secretome did not reverse senescence and did not restore inflammatory secretion in cancer cells.

An important aspect in tumor progression is the migration of cancer cells from the primary location to secondary distant sites. Paracrine signaling of senescent stromal cells, which are abundant in benign tumors and are present in cancer [32,33], affects the malignant properties of tumor cells, mainly enhancing their proliferation and invasiveness [34,35,36]. To date, there is a growing body of research showing pro-tumorigenic role of SASP [37,38,39]. Conversely, information that the paracrine secretion of senescent cells can inhibit tumor development is very scarce and contradictory. In this context, studies of the senescence of hepatic stellate cells and its role in creating tumor suppressive microenvironment are of particular interest. Liver fibrosis that precedes the development of cirrhosis and liver cancer mainly arise due to activated stellate liver cells. However, when liver stellate cells undergo replicative or induced senescence, they exhibit a less fibrogenic secretory phenotype and are prone to spontaneous apoptosis [40,41]. The p53-expressing senescent stellate cells via SASP distort the polarization of macrophages toward the M1-state, inhibiting the tumor [42]. Interleukin-10 has been demonstrated to induce hepatic stellate cell senescence and alleviate liver fibrosis via STAT3-p53 pathway [43]. Lujambio et al. demonstrated that macrophages previously exposed to CM of senescent stellate cells did not affect the proliferation of precancerous hepatoblastoma progenitor cells in co-culture, while pre-exposure of macrophages with CM of proliferative stellate cells significantly increased the growth of precancerous cells [42]. Thus, there is a mechanism in the liver that prevents the development of fibrosis and cancer due to the induction of senescence in hepatic stellate cells, the secretion of which does not possess pro-oncogenic properties.

Interestingly, analysis of genes driving cellular senescence and SASP in mammals showed a close overlap with anti-longevity genes rather than genes that promote longevity [44]. In this context, the evolution of longevity should include the selection of mechanisms that provide SASP inhibition along with a non-permissive cancer microenvironment. In our previous reports, we have shown that *Spalax* resists cancer and maintains a tumor protective microenvironment [9,13]. Suppression of the inflammatory response in senescent cells and tissues seems to be one of the main antitumor strategies of *Spalax* [14].

Targeting cancer cell migration is one of the main approaches to reducing tumor progression. Here, we demonstrate that senescent secretome of *Spalax* fibroblasts induced senescence in breast carcinoma cells MDA-MB-231 and MCF-7 while inhibiting migration of cancer cells. MDA-MB-231 cells are extremely aggressive metastatic malignant cells with a high level of secretion of their own inflammatory growth factors [18]. The induction of senescence in such cells, if accompanied by the establishment of the canonical SASP, should enhance the secretion of inflammatory factors. However, we received the opposite effect, namely the secretion of SASP was repressed. Exposure to *Spalax* CM significantly reduced *IL-6* gene expression and IL-6 protein secretion in MDA-MB-231 cells. Another breast cancer cell line, MCF-7, also reduced malignancy under the influence of paracrine factors of senescent *Spalax* cells. Namely, MCF-7 cells acquired senescence and reduced the ability to migrate while reducing the expression of *IL-6*. A pleiotropic inflammatory cytokine IL-6 is considered a key factor in the growth of malignant neoplasms and metastases [45]. IL-6 facilitates epithelial to mesenchymal transition in lung cancer increasing vimentin expression [46], and in colorectal cancer via integrin β6 upregulation [45]. A growing body of reports demonstrates a strong correlation between IL6 levels and the invasiveness of breast cancer [45,47]. The small soluble protein IL8, which belongs to the CXC chemokine family, is overexpressed in breast cancer, and high IL8 levels are associated with poor prognosis and invasion [48]. GROα, ICAM 1 and Cox-2 are SASP factors that are also closely associated with the progression and metastasis of breast cancer [49,50,51]. All these factors belonging to SASP were suppressed in MDA-MB-231 by senescent secretome of *Spalax*. Since transcription factor NF-κB controls the secretion of inflammatory mediators, we analyzed the status of expression/phosphorylation of NF-κB-related proteins in MDA-MB-231 cancer cells treated with *Spalax* CM. The phospho-NF-κB antibody array showed suppression of about 60% of proteins, including: NF-κB-p65, NFkB-p105/p50, IkB α/β, CBP/p300, p38, all strongly involved in SASP secretion. Among the proteins engaged in migration, the suppression of the expression and phosphorylation of phospholipase C-gamma 1(PLCγ1) is of particular interest. PLCγ1 phosphorylation status was shown to be a prognostic marker of metastatic risk in patients with breast cancer [52]. Antibody microarray demonstrated decreased phosphorylation of PLCγ1 in different sites of tyrosine (1253,783 and 771). Inhibition of phosphorylation of PLCγ1 (tyrosine 771) has been shown to reduce the risk of brain metastasis in experimental breast cancer [53].

IL1-α is a key protein among SASP molecules and, despite its minor secretion, is a regulator of NF-κB activation, its nuclear translocation, and subsequent expression of major inflammatory cytokines [34,54]. IL-1α expression has been reported to be significantly increased in many types of human malignant neoplasms and involved in cancer progression and metastasis [55,56,57]. IL1α stimulates an inflammatory response, mainly as a surface membrane-bound protein, which binds to the IL-1R receptor, initiating a signaling cascade that leads to NF-κB activation, translocation to the nuclei and subsequent production of inflammatory factors [58]. IL1α/NF-κB positive feedback loop is disrupted in *Spalax* senescent fibroblasts as we described [14] where we found no membrane-bound IL1α in senescent *Spalax* cells, while IL1α was mainly centered around the nuclei. Like *Spalax*, IL-1α of MDA-MB-231 cells in response to *Spalax* CM undergoes redistribution, namely, it disappears from the membrane, which obviously disrupts signal transmission from IL1R receptor to NF-κB. Lau et al. demonstrated that breakdown of the IL-1α signaling pathway effectively suppresses SASP but does not abolish cell cycle arrest in senescent cells [59]. These authors demonstrated that genetic ablation of IL-1α decreases pancreatic cancer progression. Our findings show that *Spalax* senescent secretome abrogated SASP and inhibit breast cancer cell migration by targeting IL-1α. Our observations are consistent with the data of Lau et al. [59], as they show that inhibition of IL-1α is responsible for suppressing inflammatory secretion in senescent cells without affecting cell cycle exit. Targeting the IL-1 signaling pathway to separate SASP from cell cycle exit is currently being considered as a useful new strategy for preventing aging-related inflammaging and slowing cancer progression [60].

How to induce aging of cancer cells while suppressing the harmful effects of SASP is a question that is currently being extensively explored. In this regard, the discovery of natural, well-tuned mechanisms of suppressing the inflammatory response in *Spalax* fibroblasts, as well as the mechanisms of transmission of paracrine senescence, not associated with the inflammatory component of SASP, into cancer cells could significantly accelerate the development of a complex of artificial molecular inhibitors for modulating SASP both in cancer cells and in the senescent stromal cells of the cancer microenvironment. Our previous evidence suggests that inhibition of SASP in *Spalax* is a strategy that supports healthy aging, free of inflammation-related diseases, including cancer. In this report, we obtained evidence that *Spalax* microenvironment not only does not support tumor lesions, but also ‘educates’ cancer cells to suppress the secretion of inflammatory factors in themselves, thereby maintaining a non-permissive cancer microenvironment.

## 4. Materials and Methods

### 4.1. Cell Culture and Generation of Conditioned Medium (CM)

*Spalax* dermal fibroblasts were isolated from newborns as we described earlier [9] and stored in liquid nitrogen in our laboratory prior to use in the current work. Human foreskin fibroblasts and breast cancer cells MDA-MB-231 and MCF-7 were obtained from ATCC^®^, Manassas, VA, USA. *Spalax* and human fibroblasts were grown in DMEM-F12 medium, MDA-MB-231cancer cells were grown in DMEM high glucose medium (supplemented with 10% FBS, L-glutamine (2 mM) and penicillin-streptomycin (100 u/mL, 0.1 mg/mL respectively) in standard CO_2_ incubator. Growth media and supplements were purchased from Biological Industries (Beit Haemeq, Israel). *Spalax* fibroblasts were serially passaged to achieve replicative senescence as we described [14]. At passage 5–7, most cells acquired an enlarged, flattened morphology, cell division was reduced below 25% of the total population, and cells showed positive SA-β-Gal staining and increased expression of senescence-associated p53 and p21.

Human foreskin fibroblasts reach senescence after about 40 passages [14]. Pre-senescent human fibroblasts (passages 30–35) were stored in liquid nitrogen and thawed to bring the cells to senescence and obtain senescence secretome.

To create Spalax senescent secretomes (senescent CMs), *Spalax* senescent cells were incubated in DMEM-F12 medium supplemented with 10% FBS for 8–10 days without changing the medium. The complete supernatants were collected and centrifuged at 120× *g* for 5 min at room temperature to remove any cell debris.

### 4.2. Treatments of Cancer Cells with CMs and Evaluation of Senescence

*Spalax* and human CMs harvested from senescent fibroblasts were tested for anti-cancer activity by measuring cancer cell viability using the PrestoBlue^®^ reagent (TermoFisher Scientific, Waltham, MA USA) [9]. Hep3B (HepG2) cells were used as reference cells due to their high sensitivity to factors secreted by *Spalax*, so the viability reduction effect could be assessed as early as 3–4 days. MDA-MB-231 and MCF-7 cells were exposed to CMs of senescent *Spalax* or human fibroblasts for various periods of time depending on the experiment. CM was added in a 1:1 ratio with fresh culture medium. With prolonged exposure (more than 5 days), CM was replaced by the same CM 1:1 with fresh medium.

To investigate the role of IL-1α, recombinant human IL-1α protein (IL-1α agonist, Abcam, Cambridge, United Kingdom) was added to MDA-MB-231 (50 ng/mL) along with treatment with CMs.

#### 4.2.1. Senescence-Associated β-Galactosidase (SA-β-Gal) Staining

Cancer cells were seeded at 2.5 × 10^4^ cells/well in six-well plates and treated by either *Spalax* or Human CMs. After 96 h (or after 10-day) of treatments, SA-β-Gal activity was determined. The X-Gal stock solution was prepared by dissolving 40 mg/mL X-Gal (Invitrogen, Carlsbad, CA, USA) in dimethylformamide immediately before staining. SA-β-Gal staining solution was prepared as follows: 1 mg/mL of X-Gal stock solution was dissolved in phosphate buffered saline containing 5 mM potassium ferrocyanide, 5 mM potassium ferricyanide, 2 mM MgCl_2_, adjusted pH to 6.0. Cells were fixed using 0.2% glutaraldehyde in PBS for 15 min at RT, washed in PBS and incubated in fresh SA-β-Gal staining solution for overnight at 37 °C. The cells were checked for development of the blue color under a light microscope. Quantitative analysis was performed using Image J software in four independent fields (in triplicates).

#### 4.2.2. Population Doubling

Population doubling levels of MDA-MB-231 cells were assessed as follows: 2.5 × 10^4^ cells were plated in six-well plates and treated with CMs for 24, 48, 72, and 96 h, cells were trypsinized and counted in triplicates. For long-term treatment (10 days), MDA-MB-231 cells were seeded in 12-well plates at a density of 0.5 × 10^4^ cells per well. *Spalax* CMs were added in ratio 1:1 with fresh culture medium. The cell number was counted after 5, 7 and 10 days. CM was replaced after 5 days of treatment with the same CM in a ratio of 1:1 with fresh nutrient media.

#### 4.2.3. Cell Cycle Analysis

MDA-MB-231 cells were seeded in six-well plates, treated with CMs or untreated (5 × 10⁵ cells for each probe), washed twice with PBS, trypsinized, and transferred to 5- mL tubes, then washed three times with PBS and centrifuged at 850× *g*, thereafter hypotonic buffer (Sodium citrate 0.1%; Triton 0.1%) was added to the pellet of the cells followed by propidium iodide (PI) staining (final concentration 25 μg/mL); The PI fluorescence of individual nuclei was recorded by FACSaria (Becton Dickinson, NJ, USA). A total of 10,000 events were acquired and corrected for debris and aggregates.

### 4.3. Evaluation of Cancer Cell Migration

#### 4.3.1. Scratch Assay

MDA-MB-231 and MCF-7 cells were seeded into 12-well plates. When the cells formed a confluent monolayer (80%), the cell cultures were scratched with a sterile 200-μL tip to form a “wound” and incubated with or without CMs. The migration of cancer cell across the scratch wound was monitored (0, 12, and 24 h). The cells were photographed under phase-contrast microscopy. Scratch wound gaps were measured using Image J software.

#### 4.3.2. Transwell^®^ Migration Assay

CMs of *Spalax* and human fibroblasts were added to the lower chamber (1:1 with fresh DMEM-F12 medium supplemented with 10% FBS); suspension of MDA-MB231cells (1 × 10^5^) in 500 μL of DMEM containing 10% FBS were added to the upper chamber. After 24 h membranes of the inserts were fixed with 2.5% glutaraldehyde solution for 10 min, washed with DDW, and stained with 0.5% Toluidine Blue for 5 min. Migrated MDA-MB-231 (adhered to the lower surface of the transwell membranes with an 8 μm-pore size) were photographed. Migrated cancer cells were calculated using Image J software in four independent fields.

### 4.4. Immunoblotting

Following the treatments, cells were washed with ice-cold PBS, lysed in RIPA/SDS buffer containing sodium orthovanadate phosphatase inhibitor and Complete Protease Inhibitor (Roche Diagnostics GmbH Roche Applied Science, Mannheim Germany). Samples were centrifuged, and supernatants were collected. Protein concentrations were determined with Bradford Assay (Bio-Rad Laboratories, Hercules, CA, USA); equal concentrations of proteins were then electrophoresed, blotted onto nitrocellulose membrane and incubated with primary antibodies over night at 4 °C, thereafter membranes were washed and incubated with secondary antibodies for 1 h at room temperature. Protein bands were visualized by a chemiluminescence detection kit for HRP EZ-ECL (Biological Industries, Beit Haemek, Israel) using MyECL Imager (Thermo Scientific, Wohlen, Switzerland) and quantified by Quantity One^®^ 1-D analysis software (Bio-Rad Laboratories, Hercules, CA, USA). Information about antibodies used for the western blot analysis is presented in Appendix A.

### 4.5. Immunofluorescence

MDA-MB-231 cells were seeded in 6-well plate on glass coverslips at ~40 k cells/well and treated with *Spalax* and human CM or untreated. Thereafter, cells were fixed at room temperature with chilled methanol (−20 °C) for 5 min and washed twice with ice-cold PBS. After permeabilization for 10 min with PBS containing 0.5% Triton X-100, 1% tween, 0.1% bovine serum albumin (BSA) cells were blocked with 1% in PBS, cells were processed for immunostaining with primary/secondary antibodies. To evaluate the level of DSBs cells were stained with γ-H2AX antibodies and counterstained with DAPI as we described [12]. Cells were visualized under fluorescent microscope (Leica DMi8, equipped with Leica DFC365FX camera). The images were used for quantification of the foci using FociCounter software. At least 250 nuclei from several random fields were scored. For p53 visualization (1026-1) p53 RabMAb^®^ (primary Ab) and anti-rabbit Alexa Flour 647 (secondary Ab), were used. The preparation of cells for staining with NF-κB-p65 and IL-1α was performed in the same manner as described above. Specification of antibodies and dilutions are shown in Appendix A.

### 4.6. Preparation of RNA, cDNA

Total RNA from freshly washed cells were extracted using RNeasy Mini Kit (QIAGEN) following the manufacturer’s instructions. cDNA samples were synthesized using iScript™ cDNA Synthesis Kit (Bio-Rad Laboratories Life Science Group, Hercules, CA, USA).

### 4.7. Quantitative Real-Time Polymerase Chain Reaction (RT-PCR)

Species-specific primers were designed for each target by using Primer3 software (Applied BioSystems, San Francisco, CA, USA) based on the published sequences. Relative quantification of gene transcription was performed by using Fast SYBR Green (Applied BioSystems, San Francisco, CA, USA), and 1 μL of cDNA generated from 50 ng total RNA. Serial dilutions of the cDNA with the highest expression level for each target gene were used to build a relative standard curve and to test amplification efficiency for each experiment. Samples were tested in triplicates. The amplification parameters were as follows: 95 °C for 20 s, followed by 40 cycles of 95 °C for 3 s and 60 °C for 30 s. To verify a single product with fixed melting temperature, melting curve protocol was applied. The quantification relied on equal amounts of total RNA used in each sample, and the reliability of this method was tested and confirmed by housekeeping genes (Appendix A). Primers are presented in Appendix A.

### 4.8. Antibody Array

An antibody microarray for phospho-NF-κB PNK215 (Fullmoon, Biosystems, Sunnyvale, CA, USA) was performed using protein lysates from MDA-MB-231 cells, either untreated or treated with *Spalax* CM. This array consists of 215 highly specific antibodies designed to profile various proteins and specific phosphorylation sites involved in signaling pathways associated with NF-κB and inflammation. Cell preparation, protein determination, dye labeling, and embedding into slides containing antibodies were performed according to the manufacturer’s instruction. To obtain spots-images, we used GenePix^®^ Microarray Scanner (Molecular Devices, San Jose, CA, USA). Data were analyzed using Protein Array Analyzer for Image J software.

### 4.9. Enzyme-Linked Immunosorbent Assay (ELISA)

Human IL-6 and IL-8 ELISA kits (R&D systems, Minneapolis, USA) were used, according to the instructions of manufacturer. Briefly, cancer cells were treated with CMs for 4 days, then cells were washed twice with PBSx1 and incubated in DMEM serum-free media for 24 h. Complete supernatant was collected, (centrifuged at 120× *g* for 5 min at RT) and processed for ELISA.

### 4.10. Statistics and Reproducibility

Experiments were repeated at least three times, unless mentioned otherwise. We used newborn fibroblasts from three different offspring of *Spalax* individuals caught at various times and in different territories of Israel. The data are presented as mean ± SD. The Mann–Whitney nonparametric test was applied to test the differences between groups; *p* < 0.05 was considered significant.

## Figures and Tables

**Figure 3 ijms-24-05132-f003:**
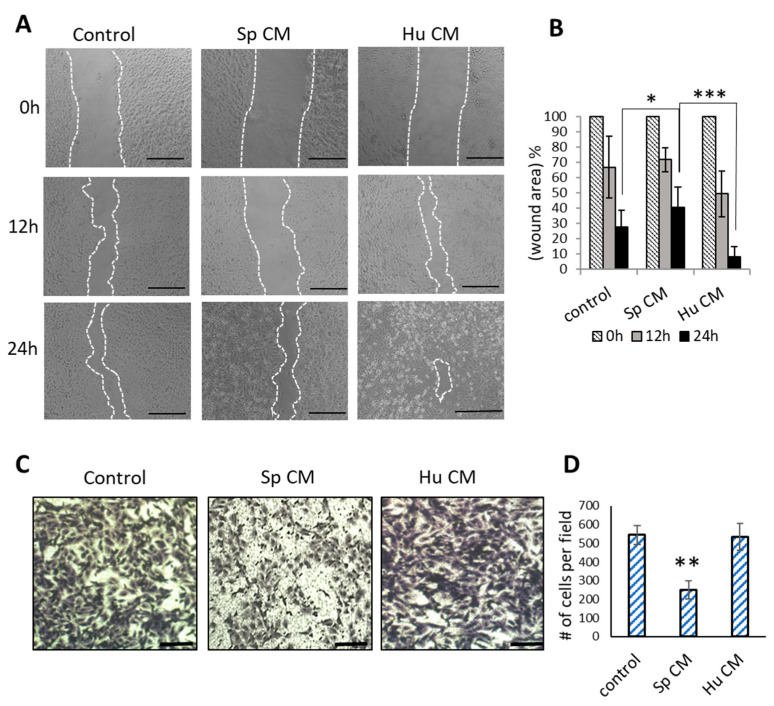
Effects of *Spalax* and human CMs on MDA-MB-231 migration (scratch assay and Transwell^®^ migration assays). (**A**) Representative images of wound healing in MDA-MB-231 cells treated with *Spalax* (Sp) and human (Hu) CMs. Cell monolayers were scratched with sterile pipette tip to form wounds. The scratch wounds were monitored after 12- and 24-h. Bars, 200 μm (**B**) Percentages of cell free area were calculated and are presented as mean ± SD of three independent experiments (n = 3), * *p* ≤ 0.05 differences between wound area of *Spalax* CM and control at 24 h. *** *p* ≤ 0.05 differences between wound area of *Spalax* CM and human CM at 24 h (**C**) Transwell^®^ migration assays: CMs of *Spalax* and human fibroblasts were added to the lower chamber; suspension of MDA-MB231cells (1 × 10^5^) in 500 μL of RPMI were added to the upper chamber. Migrated MDA-MB-231 (adhered to the lower surface of the transwell membranes with an 8 μm-pore size) were photographed after incubation for 24 h at 37 °C. Representative images are shown. Bars, 100 μm (**D**) Migrated cells were calculated using ImageJ software in four independent view fields per insert. Experiments were performed in triplicates and repeated 3 times. Data are presented as mean ± SD (n = 3), ** *p* ≤ 0.001 differences between number of migrated MDA-MB-231 under effect of *Spalax* CM versus control. # means the number.

**Figure 4 ijms-24-05132-f004:**
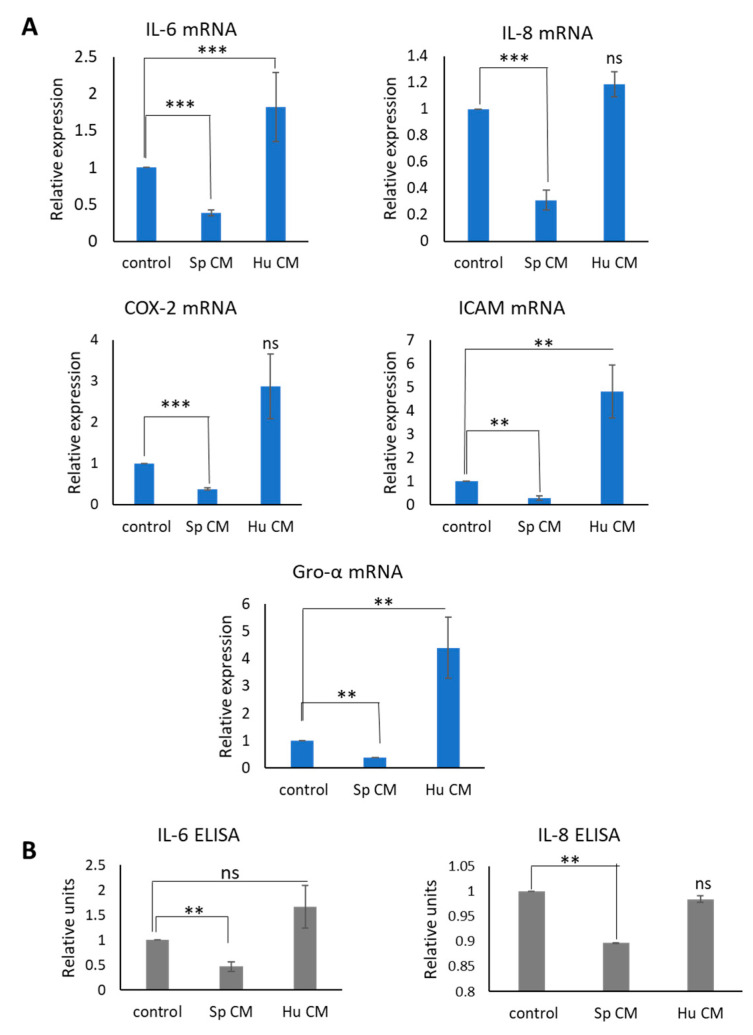
The expression and secretion of inflammatory mediators in MDA-MB-231 cells under exposure to *Spalax* (Sp) or human (Hu) CMs. (**A**) The mRNA expression rates were quantified by using qRT-PCR. Representative data for *COX-2 IL8, IL6* and *GRO-α* and *ICAM-1* are shown. Experiments were performed in triplicates and repeated at least three times. Data are presented as mean ± SE; ** *p* ≤ 0.01; *** *p* ≤ 0.001, differences between treated MDA-MB-231 cells vs. untreated; “ns”, not significant *p* value (**B**) MDA-MB231 were incubated for 4 days with CM from senescent *Spalax* and human fibroblasts, thereafter cells were washed and incubated for further 24 h in fresh serum-free media for ELISA analysis. Experiments repeated at least three times with CMs received from different *Spalax* individuals and from different late passage- CMs of human fibroblasts.

**Figure 7 ijms-24-05132-f007:**
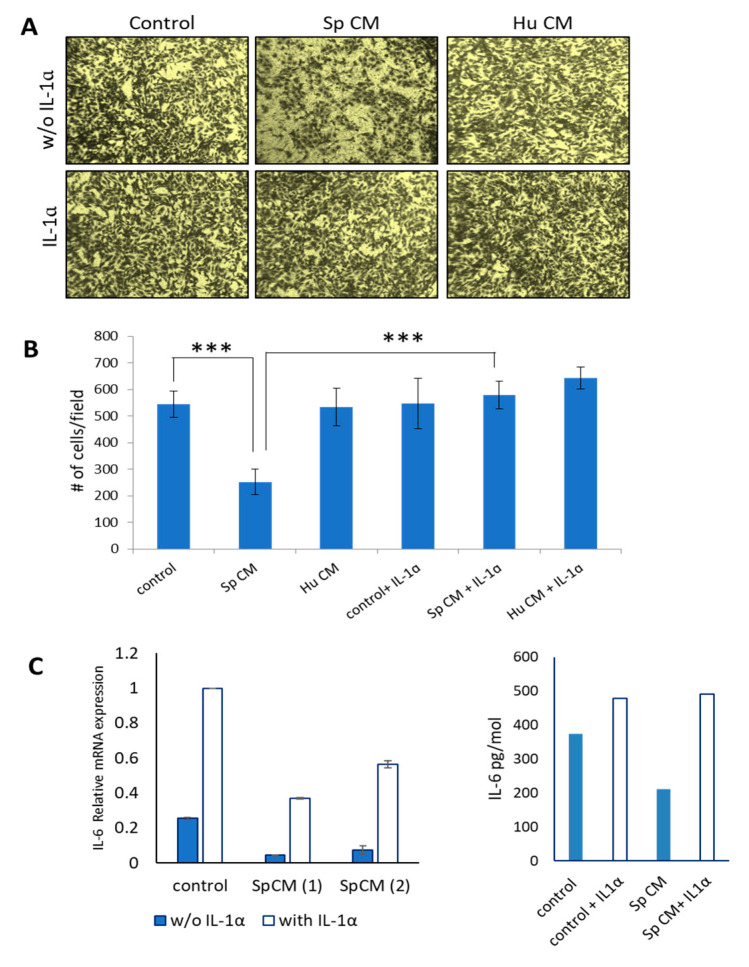
Recombinant IL1α agonist restores cancer cell migration when added to MDA-MB-231 cells exposed to *Spalax* CM. (**A**) Transwell^®^ migration assays: CMs of *Spalax* (Sp) and human (Hu) fibroblasts with/without of 50 ng/mL recombinant IL-1α were added to the lower chamber; suspension of MDA-MB231cells (1 × 10^5^) in 500 μL of RPMI were added to the upper chamber. Migrated MDA-MB-231 (adhered to the lower surface of the Transwell^®^ membranes with an 8 μm-pore size) were photographed after incubation for 24 h at 37 °C. (**B**) Migrated cells were calculated using Image J software in four independent view fields per insert. Experiments were performed in triplicate and repeated 3 times. Data are presented as mean ± SD (n = 3), *** *p* ≤ 0.001 differences between number of migrated MDA-MB-231 under effects of *Spalax* CM versus control, and between effects of SpCM+IL-1α versus SpCM alone. (**C**) MDA-MB-231 were incubated for 4 days with CM from senescent *Spalax* fibroblasts with/without recombinant IL-1α (50 ng/mL), thereafter cells were either collected for mRNA determination or washed and incubated for further 24 h in fresh serum-free media, then the IL6 ELISA was applied. *Spalax*. CMs (1) and (2) are the senescent secretome received from fibroblasts isolated from different *Spalax* individuals. # means the number.

**Figure 8 ijms-24-05132-f008:**
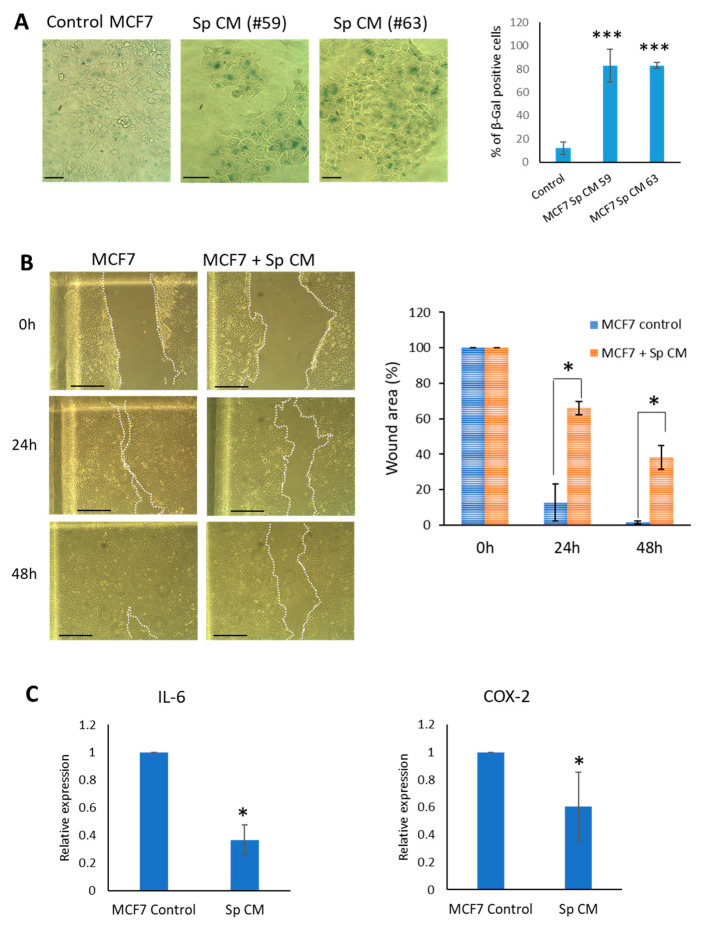
Influence of *Spalax* CM on the level of SA-β-Gal activity, migration and expression of SASP factors in MCF-7 breast cancer cells (**A**) SA-β-Gal staining representative images. MCF-7 cells were treated with *Spalax* CMs for 4 days. The percentage of SA-β-Gal-positive cells calculated from at least 300 cells in four independent fields for each biological repeat (n = 2) in triplicate (*Spalax* CMs were collected from senescent cells of two independent individuals (#59 and #63). Data presented as mean ± SD. *** *p* < 0.001; differences between control MCF-7 cells (untreated) and treated with *Spalax* CMs; (**B**) Scratch assay. Representative images of wound healing in MCF-7 cells untreated or treated with *Spalax* (Sp) CM. Cell monolayers were scratched with sterile pipette tip to form wounds. The scratch wounds were monitored after 24 and 48 hrs. (**B**) Percentages of cell free area were calculated and are presented as mean ± SD of two independent experiments (n = 2), * *p* ≤ 0.05 differences between wound area of MCF-7 cells exposed to *Spalax* CM and control at 24 h/48 h. (**C**) The expression of SASP genes *IL-6* and *Cox-2* in MCF-7 cells after exposure to Spalax CM. The mRNA expression rates were quantified by using qRT-PCR. Data presented as mean ± SD. * *p* < 0.05; differences between control cells (untreated) and treated with *Spalax* CM.

## Data Availability

The authors declare that all the data supporting the findings of this study are available in the article and its Appendix A.

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
