# Peer review of "Senescent Secretome of Blind Mole Rat Spalax Inhibits Malignant Behavior of Human Breast Cancer Cells Triggering Bystander Senescence and Targeting Inflammatory Response"

_ijms, 2023, doi:10.3390/ijms24065132_

Round 1

Reviewer 1 Report (New Reviewer)

After reading and rereading this manuscript, which stems from intensive laboratory work, I really have few comments or criticisms to add. In addition, the Academic Editor has already made wise comments and suggestions that the authors have already answered and incorporated into the manuscript properly. I want to congratulate the authors for the extensive and valuable work they have done.
Given the extent of the laboratory work and the consequent intensity of the manuscript, it seems appropriate that Results should be lightened, for example the small "introductions" and References (in Results) could be removed.
As a general consideration, I think that the authors could summarize, and not repeat certain concepts or explanations that are repetitive in the context of a manuscript that is very extensive.

Author Response

We are very grateful to the Reviewer for carefully reading the manuscript, as well as for the high appreciation of our work after a major revision. Based on your recommendation, we have revised the results and have either deleted the duplicates or moved them to a discussion. Most of the references have been omitted from the results, excluding of those we deemed necessary to retain consistency.

Reviewer 2 Report (New Reviewer)

This study is well conducted and the results are of high quality convincingly showing that Spalax secretome is inducing some features of senescence in MDA-MB-231 cancer cells but without the canonical inflammatory response.

I would however suggest some points to be improved prior to publication in IJMS

-        - The RT-qPCR data (Figures 2 and 4) should include the data presently shown as supplementary data in order to be able to perform statistical analysis.

-        - The graph shown on Fig. 5B should be better explained in the text.

-       -  It is not clear neither in the core of the manuscript nor in the Materials and Methods part whether the authors used IL1α or an agonist of IL1α. This should be clarified

-        - Figure 7C : in the legend, it is written that CM from human fibroblasts has been used, but the corresponding data is not shown on the figure. On the left graph, what are SpCM (1) and SpCM (2)? This should be clarified

-        In the discussion part, the authors argue that they have shown that Spalax CM is capable of transmitting a senescent phenotype to cancer cells and a suppression of metastatic behavior of these cells (line 345). Although, the authors indeed showed a decrease of features associated with metastasis such as a decrease migration of the cells, they do not present data in vivo formally demonstrating a reduced metastatic potential of these cells. Therefore, I suggest that the authors should be more careful in their conclusions.

Author Response

We thank the Reviewer for carefully reading our original submission and for making suggestions that helped us greatly to improve the quality of our manuscript. Below you can read our answers to specific questions:

       - The RT-qPCR data (Figures 2 and 4) should include the data presently shown as supplementary data in order to be able to perform statistical analysis.

Response: We revised figures 2 and 4 and combined the data presented in the main figures with the data of appendices. The statistics are now presented in new figures 2 and 4.

      - The graph shown on Fig. 5B should be better explained in the text.

Response: Explanation is now incorporated (page 8) "Volcano plot (Figure 5B) displays the statistical significance of the differences (relative to the magnitude of difference) for every individual protein in the groups of MDA-MB-231 cells treated with Spalax CM versus untreated. Most NF-κB related proteins are significantly downregulated in treated cells versus control. Data points closer to "0" representing proteins that have similar expression levels (empty points)".

     -  It is not clear neither in the core of the manuscript nor in the Materials and Methods part whether the authors used IL1α or an agonist of IL1α. This should be clarified

Response:  We indicated that an IL-1α agonist was used for the study and point it in Materials and Methods (page 16) and throughout the text.

       - Figure 7C: in the legend, it is written that CM from human fibroblasts has been used, but the corresponding data is not shown on the figure. On the left graph, what are SpCM (1) and SpCM (2)? This should be clarified

Response: in this experiment, we did not use human CM, and this error was corrected in the legend for Figure 7C. We have also explained what CM 1 and 2 are.

-        In the discussion part, the authors argue that they have shown that Spalax CM is capable of transmitting a senescent phenotype to cancer cells and a suppression of metastatic behavior of these cells (line 345). Although, the authors indeed showed a decrease of features associated with metastasis such as a decrease migration of the cells, they do not present data in vivo formally demonstrating a reduced metastatic potential of these cells. Therefore, I suggest that the authors should be more careful in their conclusions.

Response: We thank the Reviewer for raising this challenge. Now we have formulated this conclusion more carefully:  "Having become senescent, breast cancer cells MDA-MB-231 and MCF-7 replicate the SASP pattern of senescent Spalax cells, namely, they demonstrate a decrease in the secretion of inflammatory mediators, which, in turn, leads to the suppression of cancer cell migration (page 12)

Reviewer 3 Report (New Reviewer)

In this study, the authors demonstrated that conditioned medium (CM) from senescent Spalax fibroblasts can transmit the senescent phenotype to human breast cancer cells without inducing an inflammatory response, thereby suppressing malignant behavior. In general, this topic is interesting for a broad readership, dealing with the mechanisms of tumor cells in response to paracrine factors of senescent microenvironment. This manuscript is well organized, and I only have minor concerns.

1.    In this study, MCF-7 breast cancer cells were used to verify the function of Spalax CM, but the relevant mechanism was not investigated. It is suggested that Figure S7 and Figure S8 are integrated into one Figure, which is placed in the text shown in Figure 8. In addition, Transwell migration assay to detect the effect of Spalax CM on MCF-7 migration was not performed.

2.    How to treat the cancer cells with CM should be described in detail in Materials and Methods.

3.    In this manuscript, all the statistical analysis charts are not standard. For example, the horizontal and vertical axis labels are inconsistent.

4.    There was no bar in Figure 1C, 2C, 3A, 3C, S7 and S8A.

Author Response

We thank the Reviewer for his time and effort he put into reviewing our report, as well as for useful comments and valuable suggestions. Below we address point-by point specific concerns:

  1. In this study, MCF-7 breast cancer cells were used to verify the function of Spalax CM, but the relevant mechanism was not investigated. It is suggested that Figure S7 and Figure S8 are integrated into one Figure, which is placed in the text shown in Figure 8. In addition, Transwell migration assay to detect the effect of Spalax CM on MCF-7 migration was not performed.

Response: In this report, MCF-7 cells were applied to confirm the data we obtained using the main model, namely MDA-MB-231. Using MCF-7 cells, we proved that the acquisition of paracrine senescence without SASP in response to Spalax senescent secretome is not a unique phenomenon that is exclusive to MDA-MB-231. However, we agreed that the results that were presented in the Figures S7 and S8 are quite important, and according to your suggestion we combined these figures to present them in the main text (Figure 8).

  1. How to treat the cancer cells with CM should be described in detail in Materials and Methods.

Response: In M&M, we have added a detailed description of how we obtained the conditioning medium from senescent fibroblasts and how we treated cancer cells with these CMs.

  1. In this manuscript, all the statistical analysis charts are not standard. For example, the horizontal and vertical axis labels are inconsistent.

Response: We have now reformatted the chart axes and statistics so that they are presented in a consistent format.

  1. There was no bar in Figure 1C, 2C, 3A, 3C, S7 and S8A.

Response: Bars are now presented in Figures 1C, 2C, 3A, 3C, 8A and B (S7 and S8A in previous version)

This manuscript is a resubmission of an earlier submission. The following is a list of the peer review reports and author responses from that submission.

Round 1

Reviewer 1 Report

I feel the manuscript should be strengthened by adding some mechanistic studies. For example, authors should identify the cytokines/components in the sp CM that mediating the observed phenotypes, instead of finishing their observation with crude condition medium—which have many components and their ratio are keep changing from time to time when the cell cycle status changed. In addition, why does the authors choice the NK-KB network antibody array instead of simply using the regular RNA sequencing to identify the altered pathways (unbiasedly)? Authors should also verify their findings (at least the top hits) from antibody arrays using individual western blot. Similarly, in Figure 1a and b, the upregulation of P53 and p21 is quite minor. Authors should validate these findings with western blot as well.

Figure 2c, the result here is against the authors’ hypothesis: gamma H2AX is a major marker of cell senescence beside the SA-β-Gal staining (while the authors did see the increase of SA-β-Gal in sp CM, Figure 1c). 

Reviewer 2 Report

The manuscript entitled "Senescent secretome of blind mole rat Spalax inhibits malignant behavior of human breast cancer cells triggering by-stander senescence and targeting inflammatory response" is an in vitro study built upon the previous finding of the authors that Spalax carmeli fibroblasts inhibit cancer cell growth (ref. 9, Manov et al., 2013). In the present manuscript the authors study the effects of Spalax fibroblasts' secretome (i.e. medium conditioned by these cells) on MDA-MB-231 breast cancer cells, using human foreskin fibroblasts' secretome as a control. While this is a brilliant idea, there are some issues with the research design that may undermine the strength of the reported findings:

1) The secretomes were collected by the authors in the presence of 10% FBS, i.e. they contain an enormous amount of growth factors, cytokines etc. already present in FBS. Especially regarding IL-1α, which is a more specific target of this research, its presence in FBS was known years ago (Hida et al., 1995, PMID: 7578990). I understand that it may be difficult to maintain the cells for 10 days in the complete absence of serum, however the authors have to determine a much lower FBS concentration that maintains the fibroblasts' viability and use it for conditioned medium production (e.g. 0.2% see Kim et al., 2018, PMID: 29967290). This comment is not overlooking the differences between the Spalax and the human fibroblast secretomes, which have been collected under identical conditions; in fact, in my view, using a lower FBS concentration may actually enhance the differences observed by the authors

2) In their previous study (ref. 9, Manov et al., 2013) the authors have used the conditioned medium diluted 1:1 with fresh one, during the assessment of its effects on other cells, and indeed this is the common practice in most labs. However, it seems that in the present study the authors used the conditioned medium as it is. Given that this medium was incubated with the cells for 8-10 days (without change, lines 465-466), how can they be sure that it was not depleted of its nutrients and other necessary constituents ? Moreover, the authors are not giving any details on how did they normalize the effects of the two secretomes from the two different species (Spalax vs. human). Did they collect medium from equal cell numbers ? Did they test the total protein content of the two secretomes ?

3) In the legend of Figure 1, the authors state that regarding Spalax CMs, triplicates were collected from senescent cells of three independent individuals, while for human CMs they were obtained from the same cells thawed at different times and at different passages (lines 149-152). This is clearly an uneven approach of the two samples. The authors have to collect CMs from three different human foreskin fibroblast strains being at an equal passage level. Actually the CMs from different passages may explain the relatively high SD (error bar) of the Hu CM sample.

4) A very important marker of senescence is the expression of p16INK4A, however this gene is not expressed in MDA-MB-231 cells. The authors are advised to test their results also in a cancer cell line expressing p16INK4A, e.g. the breast cancer MDA-MB-468 cells.

5) In Figure 6B, what is the difference between "Negative control" and "MB-231 control" ? If the second one includes the addition of IL-1α, then this should be clearly written.

There are some minor spelling mistakes, e.g.

line 131 should read "develop" instead of "develops"

line 244 obviously an "In" is missing

line 302 should read "their" instead of "its"

line 329 obviously a "was" is missing in the beginning of the line

Round 2

Reviewer 1 Report

Authors did not fully address my concerns.

Author Response

Reviewer: Authors did not fully address my concerns.

Response: We carefully addressed all comments and suggestions of Reviewer 1 in our response to Round 1 and made several changes and additions to the manuscript; nonetheless, the score of the Reviewer has not changed. The Reviewer did not specify any remarks in Round 2, while at the same time, the Reviewer had serious concerns about the consistency of the conclusions and results (according to reviewer's assessment "must be improved"). We cannot agree with this concern, since the conclusions are based on a fairly large database of statistically significant data, using independent biological repeats and a large set of adequate methods. The required by the Reviewer identification of the secreted factor is, in our opinion, not reasonable at this stage of the work; moreover, the used NF-kB antibody array clearly yielded much specific mechanistic data than one may expect from any RNA-sequencing.

Reviewer 2 Report

Given the unique behavior of Spalax fibroblasts, and based on the long experience of the authors with this species, I am accepting that these cells may secrete anticancer factors only in the presence of 10% FBS. Hence, I consider that the authors have provided satisfactory explanations to all my points, and I would recommend acceptance of the manuscript. A minor comment regards lines 441-442 of the revised manuscript, where the correct expression should be "...are undergoing senescence" (omitting "to"). 

Author Response

Reviewer: Given the unique behavior of Spalax fibroblasts, and based on the long experience of the authors with this species, I am accepting that these cells may secrete anticancer factors only in the presence of 10% FBS. Hence, I consider that the authors have provided satisfactory explanations to all my points, and I would recommend acceptance of the manuscript. A minor comment regards lines 441-442 of the revised manuscript, where the correct expression should be "...are undergoing senescence" (omitting "to"). 

Response:

We once again thank Reviewer 1 for his extremely helpful comments and suggestion.

The minor comment was fixed (line 441)